# Surface oxidation/spin state determines oxygen evolution reaction activity of cobalt-based catalysts in acidic environment

Jinzhen Huang [1] ✉, Camelia Nicoleta Borca [2], Thomas Huthwelker[2], Nur Sena Yüzbasi[3], Dominika Baster[1], Mario El Kazzi [1], Christof W. Schneider [4], Thomas J. Schmidt [1,5] & Emiliana Fabbri [1] ✉

Co-based catalysts are promising candidates to replace Ir/Ru-based oxides for oxygen evolution reaction (OER) catalysis in an acidic environment. However, both the reaction mechanism and the active species under acidic conditions remain unclear. In this study, by combining surface-sensitive soft X-ray absorption spectroscopy characterization with electrochemical analysis, we discover that the acidic OER activity of Co-based catalysts are determined by their surface oxidation/spin state. Surfaces composed of only high-spin $Co^{II}$ are found to be not active due to their unfavorable water dissociation to form $Co^{III}$-OH species. By contrast, the presence of low-spin $Co^{III}$ is essential, as it promotes surface reconstruction of Co oxides and, hence, OER catalysis. The correlation between OER activity and Co oxidation/spin state signifies a breakthrough in defining the structure-activity relationship of Co-based catalysts for acidic OER, though, interestingly, such a relationship does not hold in alkaline and neutral environments. These findings not only help to design efficient acidic OER catalysts, but also deepen the understanding of the reaction mechanism.

Proton exchange membrane (PEM) electrolyzers are promising devices to convert renewable electricity into hydrogen[1]. However, the need for noble metal catalysts, such as Ir-/Ru-based oxides, to catalyze the oxygen evolution reaction (OER) at the anodic electrode results in high capital costs[2–4]. In addition, the low abundance of Ir- and Ru-based oxides limits the scale-up application of PEM electrolyzers. Diversifying the catalyst components by utilizing catalytically active, earth-abundant elements can mitigate the reliance on Ir-/Ru-based materials and potentially reduce the capital cost and scale-up limitations of PEM electrolyzers.

Recently, Co-based materials have been extensively investigated as potential OER catalysts in acidic environments[5–12]. Crystalline cobalt spinel oxide ($Co_3O_4$) can drive the OER in strong acidic electrolytes, and has been shown to be stable for weeks under low operating

current conditions[9]. Co dissolution is inevitable under strongly acidic conditions, but the extent can be decreased with proper catalyst engineering. For example, incorporating Mn into the spinel structure can slow down the Co dissolution rate and extend the catalyst lifetime by two orders of magnitude[5]. In addition, co-doping of the Co spinel structure with La and Mn can effectively mitigate catalyst degradation during operation in a PEM electrolyzer[7]. Although Co-based oxides are not yet able to outperform Ir-/Ru-based catalysts, continuous improvements in both stability and activity demonstrate the great potential of this catalyst family for practical application in PEM electrolyzers[5,7,8].

A lack of fundamental understanding of the reaction mechanism hinders the development of more effective Co-based acidic OER

[1]Electrochemistry Laboratory, Paul Scherrer Institute, Villigen PSI, Switzerland. [2]Photon Science Division, Paul Scherrer Institute, Villigen PSI, Switzerland. [3]Laboratory for High Performance Ceramics, Empa - Swiss Federal Laboratories for Materials Science and Technology, Dübendorf, Switzerland. [4]Laboratory for Multiscale Materials Experiments, Paul Scherrer Institute, Villigen PSI, Switzerland. [5]Institute for Molecular Physical Science, ETH Zurich, Zurich, Switzerland. ✉e-mail: jinzhen.huang@psi.ch; emiliana.fabbri@psi.ch

catalysts. Based on computational studies, Li et al. suggested that the conventional adsorbate evolution mechanism, involving the formation of OOH*, is more favorable in $MnCo_2O_4$ than a mechanism involving direct O−O radical coupling[5]. In contrast, using the $^{18}O$-labeling technique, Wang et al. reported that Ba-doped $Co_3O_4$ could undergo direct O−O radical coupling during OER catalysis, rather than OOH* formation[8]. The discrepancy in the literature suggests that the understanding of the surface reconstruction and active structure of Co-based oxides in acidic environments is still insufficient.

Unlike noble metal oxides (e.g., $IrO_2$ and $RuO_2$), Co-based catalysts undergo significant surface reconstruction during the OER[13,14], which may vary depending on the surface of the pristine sample and operating conditions[6,8,15,16]. Surface reconstruction is an interfacial process driven by the interaction between the electrolyte and the catalyst surface under OER conditions. In addition to the physicochemical properties and the surface chemistry of the catalyst, the type of electrolyte has a tremendous influence on the reconstruction process and ultimately on the OER performance. In alkaline and neutral environments, the influence of the Co oxidation state and polyhedral coordination geometry on OER activity has been widely investigated. In an alkaline environment, the tetrahedral $Co^{II}$ is suggested to be responsible for the formation of active $CoOOH$[17]. In a neutral environment, Bergmann et al.[18] have proposed that the formation of active, reducible $Co^{III}$-O species at the reconstructed surface depends on the parent structure. In an acidic environment, exposing more $Co^{II}$ sites in $Co_3O_4$ has been suggested to improve the OER performance; however, the influence of the degree of crystallinity and surface area on the reported OER performance has not been investigated and may result in misleading conclusions[19]. The structure-activity relationship for OER catalysis in an acidic environment might significantly differ from that observed for Co-based catalysts in alkaline and neutral environments. Moreover, the influences of Co oxidation state (or surface $Co^{III}/Co^{II}$ ratio), and the polyhedral coordination geometry have not yet been explored for acidic environments. From this perspective, a deep understanding of the surface reconstruction mechanism and the development of structure-activity relationships for Co-based catalysts in acidic environments is crucial to elucidate the surface processes occurring under standard operating conditions, and ultimately will enable the development of novel catalyst design principles.

Based on the previous findings in literature[5,7], we have chosen CoMn-based spinel oxides as model catalysts. We tuned the $Co^{III}/Co^{II}$ surface ratio via changing the Co/Mn ratio to elucidate the active Co species and uncover the effect of the catalyst structure on OER activity in an acidic environment. In addition, to support the unveiled structure-activity relationship, six representative Co-based catalysts with surface $Co^{III}$ species (i.e., commercial $Co_3O_4$. $FeCo_2O_4$ and reconstructed $CoO$) vs. without surface $Co^{III}$ species (i.e., commercial $Co(OH)_2$, $CoCr_2O_4$ and Co-doped $CeO_2$) were investigated. This study reveals a structure-activity relationship between the OER activity of Co-based samples in an acidic environment and the Co oxidation/spin state. In particular, the pure $Co^{II}$ surface is not active towards the OER, regardless of the Co coordination environment, due to the unfavorable surface reconstruction involving the oxidation of $Co^{II}$ into $Co^{III}$ in an acidic environment. These observations differ from those reported in the literature for neutral and alkaline environments, unveiling a different structure-activity relationship for Co-based catalysts in an acidic environment.

## Results

### OER performance of $Co_xMn_{1-x}O_y$ catalysts in alkaline and acidic environment

The nano-sized $Co_xMn_{1-x}O_y$ catalysts with different nominal Co percentages (x = 1, 0.9, 0.67, 0.5, 0.33, 0.1, and 0) were prepared by flame spray synthesis (FSS)[13] followed by a thermal annealing to reduce secondary phases (see Method for more details). The dominant phase

in all the resulting $Co_xMn_{1-x}O_y$ samples is the spinel oxide, with minor secondary phases such as $CoO$, $MnO_2$, and $Mn_2O_3$ (Supplementary Fig. 1). The phase composition shifts from the $Co_3O_4$ to $Mn_3O_4$ as the Co content decreases in the sample composition. There are three different types of spinel oxides, namely normal, inverse, and mixed spinel oxides. Both $Co_3O_4$ and $Mn_3O_4$ have the normal spinel structure, while $Co_\alpha Mn_{3-\alpha}O_4$ samples ($\alpha \neq 0$ and 3) can be identified as mixed spinel oxides in which the Co partially occupies the tetrahedral site as high spin $Co^{II}$[20,21]. The overall oxidation state and spin state of Co depend on the fraction of Co in the structure, which can be varied by changing the material Co content. In addition, if Co is assumed to be the predominant active center for OER catalysis, the catalytic activity should decrease with decreasing Co content (i.e., number of active sites).

To verify this assumption, the OER activity of $Co_xMn_{1-x}O_y$ samples was evaluated in alkaline (0.1 M KOH) and acidic (0.05 M $H_2SO_4$) electrolytes (Fig. 1 and Supplementary Figs. 2 and 3). The OER polarization curves collected by chronoamperometry show a different trend in basic vs. acidic electrolytes. Generally, the OER current of $Co_xMn_{1-x}O_y$ decreases with x in an alkaline environment (Fig. 1a), supporting the widely reported hypothesis that Co is the predominant active center for the OER. In comparison, the OER current in an acidic environment shows a very sharp decrease for the samples with x < 0.9 (Fig. 1b). The corresponding Tafel plots clearly show two distinct groups among the $Co_xMn_{1-x}O_y$ samples (Fig. 1c, d). In an alkaline environment, the Tafel slope increases from ~50 to ~90 mV dec$^{-1}$ with decreasing x (Fig. 1e). In an acidic environment, the Tafel slope for each sample has a higher value than in the basic electrolyte. In particular, the lower Tafel slope of ~90 mV dec$^{-1}$ is observed for the samples with x = 1 and 0.9; a dramatic increase of the Tafel slope to above 200 mV dec$^{-1}$ is measured for the other samples with lower Co content. The extremely high values of the Tafel slope for samples with x < 0.9 suggests that these catalysts are almost inactive towards the OER in acidic environments, indicating that the primary electron transfer step is possibly turnover-limiting, with a very high symmetry factor for the OER[22,23]. The double layer capacitance ($C_{dl}$) of the catalysts was extracted to compare surface area among samples (Supplementary Figs. 4 and 5). In general, increasing the amount of Mn in $Co_xMn_{1-x}O_y$ decreases the $C_{dl}$, from ~2.48 F g$^{-1}$ for x = 1 to as low as ~0.5–1 F g$^{-1}$ for the sample with x ≤ 0.9; however, both samples with x = 1 and x = 0.9 are active in an acidic environment. Therefore, the surface area exposed to the electrolyte is not likely to be a major factor in determining the catalytic activity of $Co_xMn_{1-x}O_y$ in an acidic electrolyte. The discrepancy in the Tafel slopes in acidic vs. alkaline electrolytes supports the initial hypothesis that the OER mechanism depends on the electrocatalyst/electrolyte interfacial characteristics after surface reconstruction has occurred. The reconstruction process is controlled by both the physico-chemical properties of the pristine electrocatalyst and its interactions with electrolyte. The sudden deactivation of $Co_xMn_{1-x}O_y$ when x < 0.9 in an acidic environment could be attributed to a structural change at the Co active centers.

### Co oxidation/spin state of $Co_xMn_{1-x}O_y$ catalysts

To understand the differences in the surface chemistry and structure of the as-synthesized $Co_xMn_{1-x}O_y$ catalysts, soft X-ray adsorption spectroscopy (XAS) in total electron yield mode (TEY) with a penetration depth of around 5 nm has been used to investigate the Co and Mn L edges and the O K edge. The Co L edge spectra are sensitive to the oxidation/spin state of the Co atoms[24]. For the investigated samples, there are two major features relating to the Co $L_3$ and Co $L_2$ edges in each spectrum (Fig. 2a) because of the 2p spin-orbit coupling interaction. The Co $L_3$ edge of the pure Co sample ($Co_xMn_{1-x}O_y$, x = 1) shows typical spinel oxide features, i.e., a white line at ~780.5 eV and shoulders on the low energy side (~779 eV), correlating to $Co^{III}$ and $Co^{II}$ surface atoms, respectively[18]. The branching ratio $I(L_3)/[I(L_3) + I(L_2)]$

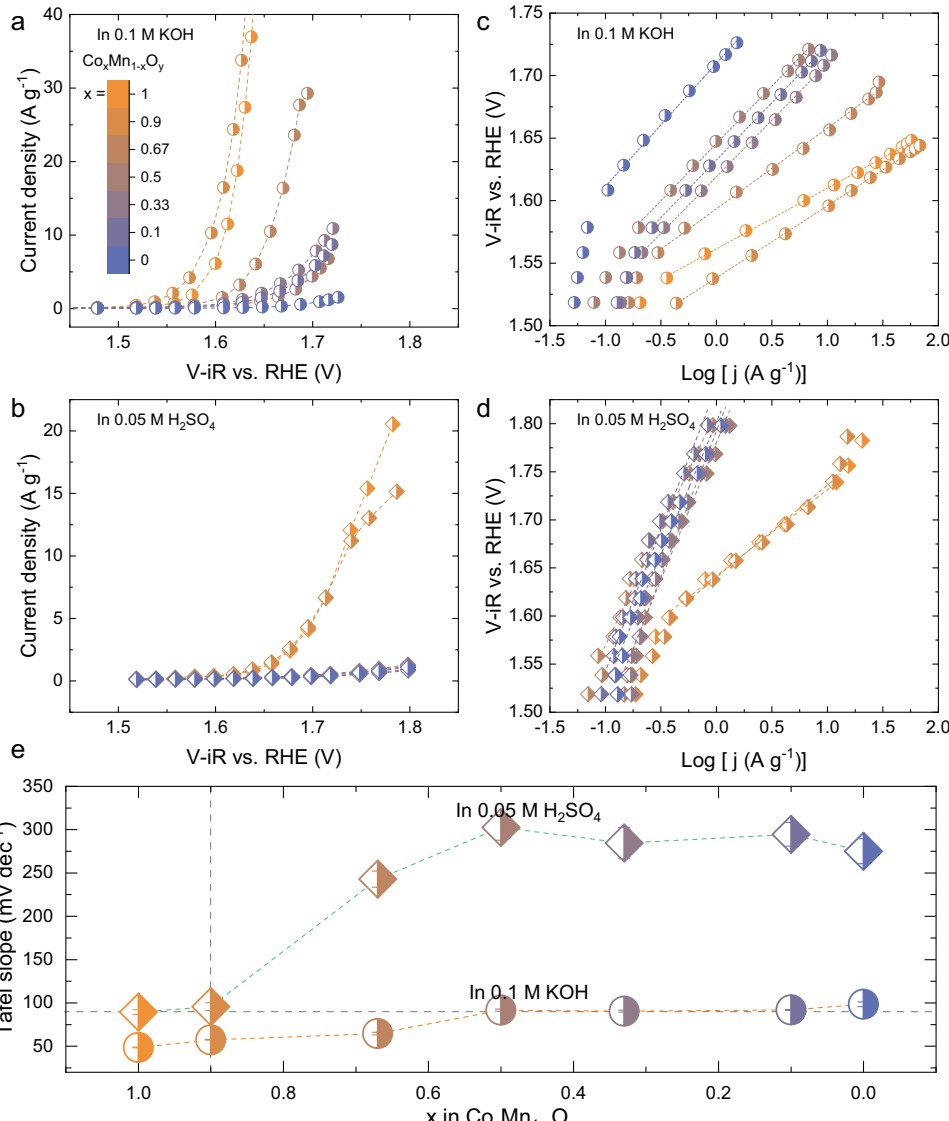

**Fig. 1 | Different OER activity of $Co_xMn_{1-x}O_y$ catalysts in alkaline and acidic environments.** Comparison of the OER activity of $Co_xMn_{1-x}O_y$ (x = 1, 0.9, 0.67, 0.5, 0.33, 0.1, and 0) in **a** alkaline and **b** acidic environments. **c, d** The corresponding Tafel plots. **e** Summary of the Tafel slopes, with error bars obtained from fitting.

can be calculated from the intensities of Co $L_3$ and $L_2$ edges (Supplementary Table 1). The representative branching ratios for low-spin and high-spin Co are 0.6 and 0.7, respectively[25]. In literature, Co atoms in $Co(OH)_2$ and CoOOH are suggested to be in high-spin and low-spin configurations[26,27], respectively, with calculated branching ratios of 0.74 and 0.61 (Supplementary Fig. 6), proving that the branching ratio is a reliable descriptor of Co spin state. For the $Co_xMn_{1-x}O_y$ series, the branching ratio increases from 0.61 to 0.69 when x decreases from 1 to 0.1. This finding indicates that the Co spin state shifts from low spin to high spin with decreasing Co content in the $Co_xMn_{1-x}O_y$ catalyst series.

Furthermore, the Co $L_3$ edge white line intensity (related to the concentration of surface $Co^{III}$ species) decreases with the Co content in $Co_xMn_{1-x}O_y$, indicating that the Co oxidation state is lower at the surface. The peak intensity ratio $I(Co^{III})/I(Co^{II})$[28]. is extrapolated from the Co $L_3$ edge to qualitatively show the changing ratio of surface $Co^{III}$ and $Co^{II}$ species, and therefore the change in surface Co oxidation state. The $I(Co^{III})/I(Co^{II})$ decreases from 2.01 for x = 1 to 0.74 for x = 0.1 (Supplementary Table 2). X-ray photoelectron spectroscopy (XPS) characterizations have been performed to further corroborate the trend revealed by TEY characterizations (Supplementary Figs. 7–9).

The deconvolution of Co 2p XPS spectra further confirms that the amount of surface $Co^{III}$ decreases with x (Supplementary Fig. 8), revealing a trend similar to the one detected by soft XAS in TEY mode. Moreover, the surface Mn oxidation state is dominated by $Mn^{II}$ and $Mn^{III}$[29,30], and the peak intensity ratio of $I(Mn^{III})/I(Mn^{II})$ also decreases with x (Supplementary Fig. 10). In line with the changes in Co and Mn oxidation state, the O K edge also shows a clear transformation as the catalyst composition changes. The peaks below and above 535 eV in the O K edge can be related to the 3d and 4sp bands of Co/Mn, respectively[25]. The peak at ~530.7 eV is associated with the Co/Mn 3d band[18], specifically the low spin $Co^{III}$-O hybridized state, and its intensity decreases rapidly when x < 0.9, accompanied by an apparent transition of the strongest peak from ~542.8 eV to 537.8 eV in the 4sp band region. At the same time, the peak intensity related to lattice oxygen in O 1s XPS spectra also shows a similar decrease when x < 0.9 (Supplementary Fig. 7b). This transition coincides with a significant structural change from $Co_3O_4$ to $Co_\alpha Mn_{3-\alpha}O_4$ ($\alpha$ = 1 or 2), as evidenced by XRD analysis (Supplementary Fig. 1). Generally, when x < 0.9 in $Co_xMn_{1-x}O_y$, this dominant phase transformation leads to significant changes in the Co, Mn and O environments at the surface, resulting in poor OER activity for these catalyts in an acidic environment.

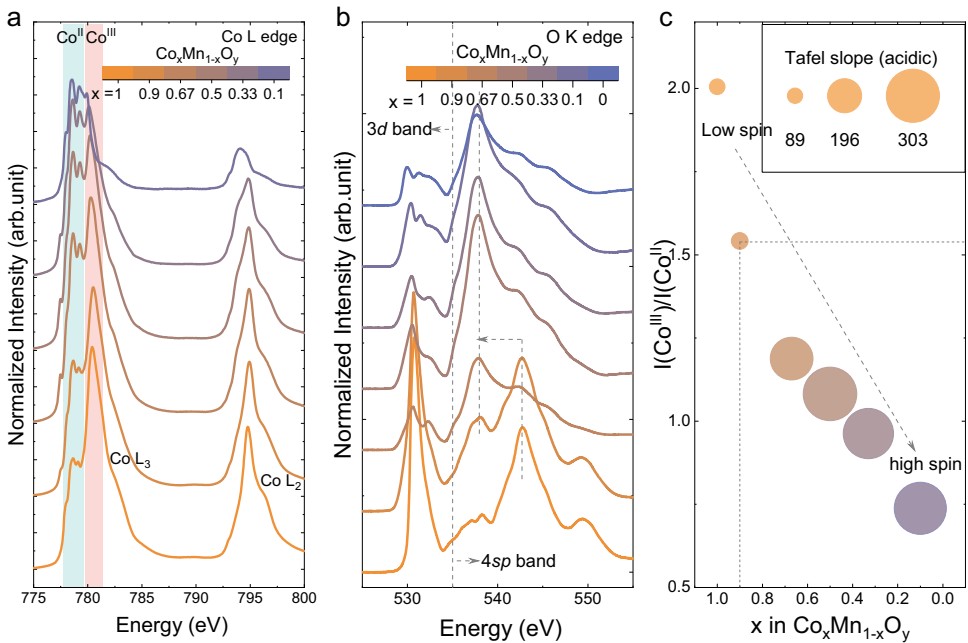

**Fig. 2 | Correlation of surface state of $Co_xMn_{1-x}O_y$ catalysts with OER activity in acidic environments. a** Co L edge and **b** O K edge of the $Co_xMn_{1-x}O_y$ samples. **c** Correlations between $I(Co^{III})/I(Co^{II})$ and Tafel slope in an acidic environment.

Furthermore, hard XAS spectra at the Co K edge were collected for $Co_xMn_{1-x}O_y$ (x = 1 and 0.67), with $CoCr_2O_4$ and commercial $Co_3O_4$ as reference samples, in which the Co oxidation state is 2+ and 2.67+, respectively (Supplementary Fig. 11). The Co K edge position ($E_{edge}$) was extracted with an integral method[6,31] to compare the Co oxidation state. The $Co_xMn_{1-x}O_y$ samples with x = 1 and x = 0.67 have higher and lower Co oxidation state than that of the commercial $Co_3O_4$, respectively. Generally, higher Co K edge position (i.e., higher Co oxidation state) is associated with higher $I(Co^{III})/I(Co^{II})$ at the Co $L_3$ edge spectra (Supplementary Fig. 11c). The lower Co oxidation state of $Co_xMn_{1-x}O_y$ with x = 0.67 compared to $Co_3O_4$, and the lower Co atomic percentage (55.5%, Supplementary Fig. 9) on the surface could account for the lower OER activity of this material compared to $CoMn_2O_4$ catalyst in literature[5]. In particular, the $I(Co^{III})/I(Co^{II})$ of $Co_xMn_{1-x}O_y$ with x = 0.67 is only ~1.19, much lower than that of commercial $Co_3O_4$ (~2.24). The in-depth surface characterization of these samples allows us to reveal groundbreaking correlations between Co oxidation state and the OER activity in an acidic environment, as shown in Fig. 2c. The Tafel slope increases significantly when the $I(Co^{III})/I(Co^{II})$ is below 1.5, i.e., when x < 0.9 in $Co_xMn_{1-x}O_y$, indicating that the OER in acidic environment is directly controlled by the surface Co oxidation state. In addition, the $Co^{III}$ is in a low-spin configuration, while the $Co^{II}$ is high spin. Therefore, the Co spin state also seems to correlate with the Co oxidation state in the $Co_\alpha Mn_{3-\alpha}O_4$ mixed spinel oxides. In summary, our results suggest that the surface dominated by high-spin $Co^{II}$ is not active towards the OER in an acidic environment.

## OER activity of catalysts with/without $Co^{III}$ on the surface

To confirm our theory, another six representative samples have been characterized. The representative samples are chosen to have different structural properties (e.g., crystal structure, doping and polyhedral coordination), and different Co surface oxidation state, which is characterized by measuring the Co L edge spectra by soft XAS in TEY mode (Fig. 3). Among the investigated samples, commercial $Co(OH)_2$, $Co-CeO_2$ (Supplementary Fig. 12) and $CoCr_2O_4$ (Supplementary Fig. 13) prepared by FSS show only the typical features of $Co^{II}$, similar to the simulated spectra of $Co^{II}$ high spin state[32]. They show no obvious peak related to the presence of $Co^{III}$ (~780.5 eV) at the surface, leading to a

very low $I(Co^{III})/I(Co^{II})$ value, but a high branching ratio (Supplementary Table 3). The Co $L_3$ edge of $Co(OH)_2$ has a typical peak at low energy (~777 eV), suggesting the presence of $Co^{II}$ atoms in octahedral coordination[33], whereas $CoCr_2O_4$ has $Co^{II}$ atoms coordinated in the tetrahedral geometry, as indicated by the absence of a low energy peak[34]. The $Co^{II}$ tetrahedral coordination in $CoCr_2O_4$ is also confirmed by hard XAS analysis at the Co K edge (Supplementary Fig. 11). The Co-$CeO_2$ show similar features to the $CoCr_2O_4$ at the Co L edge. Overall, despite the different structure, these samples only have $Co^{II}$ on the surface (hereafter referred to as $Co^{II}$-catalysts).

The commercial CoO rock salt structure is composed of $Co^{II}$ in the bulk, and its surface can be reconstructed upon contact with water during electrode preparation[33,35]. The formation of $Co^{III}$ at the CoO surface is irreversible, as confirmed by the peak intensity ratio $I(Co^{III})/I(Co^{II})$ of 1.49 (Supplementary Table 3). The $Co^{III}$ species at the surface layer were removed after sputtering with Ar-ions[36], suggesting that the bulk Co oxidation state is still 2+. The commercial $Co_3O_4$ and the $FeCo_2O_4$, produced using FSS (Supplementary Fig. 14), have a spinel structure with coexistence of $Co^{II}$ and $Co^{III}$. Therefore, these three samples will be referred to as $Co^{II}/Co^{III}$ catalysts henceforth. Interestingly, the calculated branching ratios (Supplementary Table 4) are ~0.7 (high spin) for the $Co^{II}$ catalysts, and ~0.6 (low spin) for the $Co^{II}/Co^{III}$ catalysts.

The OER performance of $Co^{II}$ vs. $Co^{II}/Co^{III}$ catalysts was evaluated, and polarization curves and Tafel plots are shown in Supplementary Fig. 15. All six representative samples show relatively high OER current in an alkaline environment; in contrast, the $Co^{II}$ catalysts are not active in an acidic environment, consistent with previous observations for $Co_xMn_{1-x}O_y$ samples. The Tafel slopes of the catalysts with low-spin $Co^{III}$ increased from ~60 mV dec$^{-1}$ in an alkaline environment to ~90 mV dec$^{-1}$ in an acidic environment (Fig. 3b), suggesting a change in the rate-determining step (RDS)[37]. The $Co^{II}$ catalysts have Tafel slopes of ~60–90 mV dec$^{-1}$ in an alkaline environment, increasing to over 200 mV dec$^{-1}$ in an acidic environment, indicating that primary electron transfer is the turnover-limiting step for the reaction[22,23]. The Co-based catalysts are more stable in an alkaline environment, as Co dissolution is inevitable under acidic conditions, though the extent is dependent on the structural properties of the catalyst and the

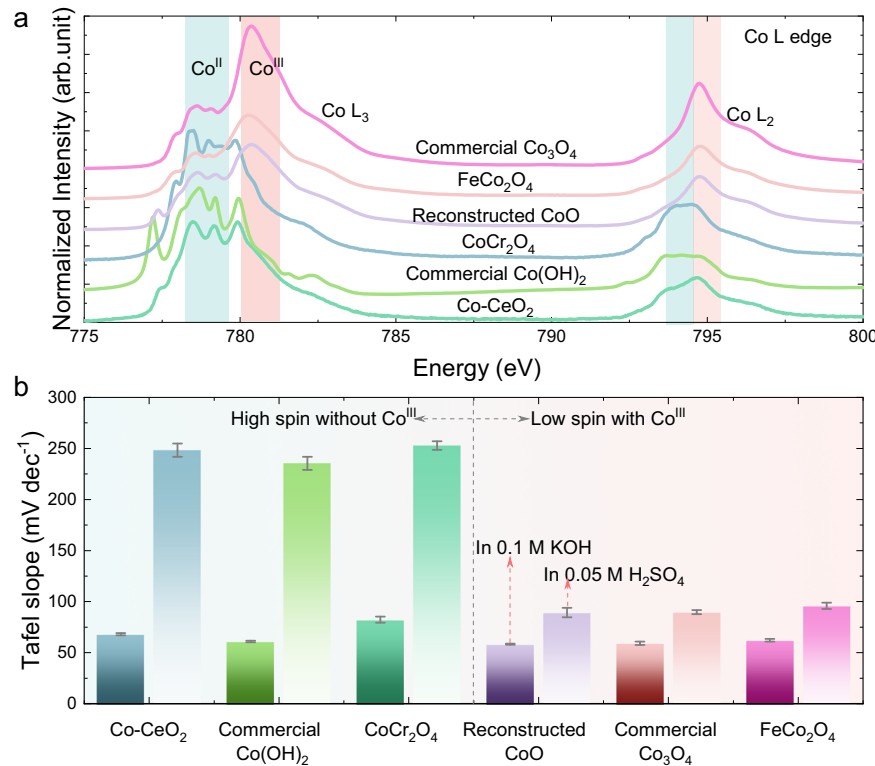

**Fig. 3 | Comparison of OER activity of catalysts with/without $Co^{III}$ on the surface. a** Different samples with dominant $Co^{II}$ or $Co^{III}$ features in the Co L edge spectra. **b** Comparison of Tafel slopes in alkaline and acidic environments, with error bars obtained from fitting.

operating conditions (e.g., applied potential, electrolyte and so on)[5,6]. We would like to point out that the poor OER activity of $Co^{II}$ catalysts is not due to significant sample dissolution during measurement. To prove this, the $Co^{II}$ catalysts were first tested in an acidic environment by performing 10 CV cycles in the OER region, showing poor activity; however, when the same (washed) electrode was immersed in an alkaline environment, high OER currents were observed in some $Co^{II}$ catalysts, indicating there is partial integrity of the electrode (Supplementary Figs. 16 and17). Taking $CoCr_2O_4$ as an example, after running 10 CV cycles in an acidic environment, it can still be activated to improve its OER performance during subsequent CV cycles in an alkaline environment (Supplementary Fig. 17a). In comparison, when $CoCr_2O_4$ is first activated to undergo surface reconstruction in an alkaline environment and then placed in an acidic environment, it shows higher OER current (Supplementary Fig. 17b), possibly due to the formation of $Co^{III}$ species during surface reconstruction under alkaline conditions[17]. Raman characterization reveals that the $CoCr_2O_4$ spinel structure is well-maintained after the OER in both alkaline and acidic electrolytes (Supplementary Fig. 18), indicating (i) the surface reconstruction in an alkaline environment is limited; (ii) the poor OER activity in acidic environments is not entirely due to catalyst dissolution.

### Correlating the Co oxidation/spin state with OER activity

To strengthen the structure-activity relationship for the OER in acidic electrolytes, the Tafel slope, Co oxidation state and Co spin state of the $Co_xMn_{1-x}O_y$ catalysts and six representative samples are summarized in Fig. 4a. The decrease in $I(Co^{III})/I(Co^{II})$ generally leads to a higher branching ratio, suggesting that the dominant Co oxidation/spin state at the surface shifts from low-spin $Co^{III}$ to high-spin $Co^{II}$. The increase in the branching ratio is more pronounced when $I(Co^{III})/I(Co^{II})$ is lower than ~1.5, and the Tafel slopes are shifted from ~90 mV dec$^{-1}$ to above 200 mV dec$^{-1}$ with the transition. Therefore, the OER activity in an acidic environment is indeed dependent on the low-spin $Co^{III}$ at the

surface. It should also be noted that some structures (e.g., rock salt CoO) have $Co^{II}$ in the bulk; however, these samples can still be active in an acidic environment if chemical reconstruction to form surface $Co^{III}$ occurs during ink preparation. Thus, the catalyst surface chemistry plays an important role in defining structure/activity relationships. Based on these results, we can conclude that the oxidation of $Co^{II}$ to $Co^{III}$ is not favorable in an acidic environment, different from the previous understanding in neutral and alkaline environments[17,18]. To support this point, the Co oxidation of commercial $Co(OH)_2$ during the OER has been tracked by the operando hard XAS characterization at the Co K edge (Supplementary Fig. 19, more details in "Methods"). The $Co(OH)_2$ is clearly active in 0.1 M KOH, but not active in 0.05 M $H_2SO_4$, as revealed by the CV profiles (Supplementary Fig. 19a). In an alkaline environment, the Co K edge position shifts to higher energy at higher applied potential, in line with increasing OER current (Supplementary Fig. 19d); in contrast, there is no obvious change in the Co K edge for $Co(OH)_2$ in an acidic environment, where the OER activity is very poor, indicating that Co cannot be oxidized in acidic conditions to promote surface reconstruction. Overall, the Co-based samples with pure high-spin $Co^{II}$ on the surface show poor OER activity in acidic environments, with no capability for surface reconstruction. In contrast, the $Co^{II}/Co^{III}$ samples can undergo surface reconstruction into CoOOH after the $Co^{II/III}$ redox peak[6], which might be further oxidized to $CoO_2$ at higher potential[16]. Therefore, we propose that the catalysts with dominant $Co^{III}$ on the surface can undergo the oxidation process of $Co^{II}Co^{III} \rightarrow Co^{III}Co^{III} \rightarrow Co^{IV}Co^{IV}$ to catalyze the OER (Fig. 4b).

We used the ethanol molecule as a chemical probe to differentiate the surface state of $Co^{II}$ catalysts vs. $Co^{II}/Co^{III}$ catalysts in an acidic environment. OH* is the first intermediate of the OER; it is electrophilic and can oxidize ethanol into aldehyde or acid[38]. Therefore, the onset of the OER polarization curve will be earlier in the electrolyte with ethanol if there are already OH* generated on the catalyst's surface; otherwise, the catalyst with weak OH* binding strength is limited by the OH* formation and shows no obvious difference between polarization

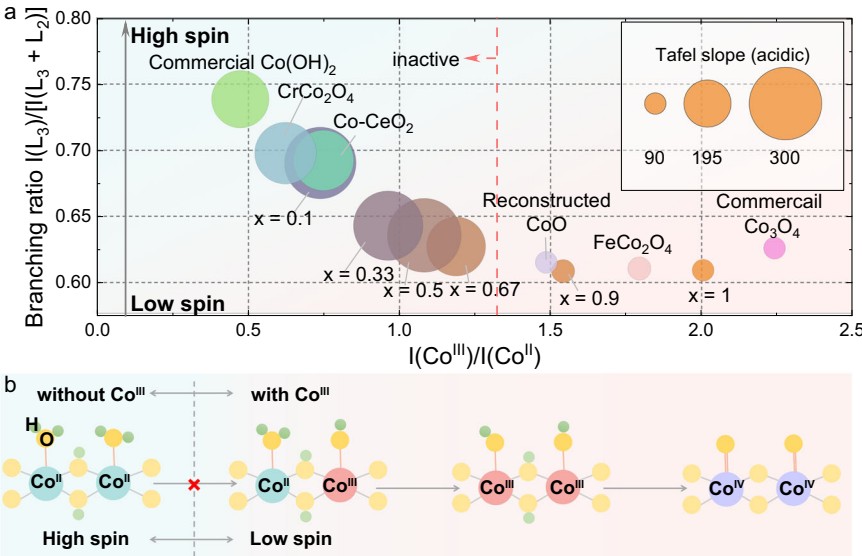

**Fig. 4 | Correlations between the Co spin/oxidation state and OER activity in acidic environments. a** The correlations between I(Co$^{III}$)/I(Co$^{II}$), branching ratio, and Tafel slope in an acidic environment. **b** Proposed surface reconstruction of Co-based catalysts with and without Co$^{III}$ for acidic OER.

curves after adding ethanol into the electrolyte. Apparently, there is no difference for the Co$^{II}$ catalysts, suggesting that the surface OH* coverage is low in acidic environments (Supplementary Fig. 20). The Co$^{II}$/Co$^{III}$ catalysts show earlier polarization onset with ethanol in the acidic environment, suggesting that the OH* formation is easier for these catalysts. Notably, the polarization onset for the Co$^{II}$ catalyst is also earlier with ethanol in the alkaline environment (Supplementary Fig. 21), indicating that the formation of OH* occurs before the onset of OER. Therefore, the OH* formation is not likely to be the RDS for all the Co$^{II}$ and Co$^{II}$/Co$^{III}$ catalysts in an alkaline environment. In an acidic environment, the RDS for Co$^{II}$/Co$^{III}$ catalysts (Tafel slope of ~90 mV dec$^{-1}$) could be the O−O bond formation[5]; the Co$^{II}$ catalysts with very poor OER activity in the acidic electrolyte (Tafel slope higher than 200 mV dec$^{-1}$) might be limited by the formation of OH*, due to the unfavorable water dissociation to form Co$^{III}$-OH species on the surface (Fig. 4b).

## Discussion

In conclusion, based on 12 different Co-based catalysts, we unveil a strong correlation between the Tafel slope and Co oxidation/spin state, thus demonstrating a different structure/activity relationship for the OER in acidic environments compared to those in alkaline environments. Compared to the bulk properties, the surface state is more critical in determining the final OER activity, highlighting the importance of surface engineering for OER in an acidic environment. We show that the surface with only high-spin Co$^{II}$ is not active towards OER in an acidic electrolyte, since the Co oxidation to promote the surface reconstruction is unfavorable; the presence of low-spin Co$^{III}$ and the ability to reconstruct into active Co species are important to initiate the OER in an acidic environment. Elucidating the active Co species for the OER in an acidic environment can provide guidance for both the design of efficient Co-based catalysts and mechanistic investigations in the future.

## Methods

### Chemicals

Co(NO$_3$)$_2$ · 6H$_2$O (98%), Ce(NO$_3$)$_3$ · 6H$_2$O (99.5%), Mn(NO$_3$)$_2$ · 6H$_2$O (98%), Fe(NO$_3$)$_3$ · 6H$_2$O (98%), CrCl$_3$ · 6H$_2$O (98%), KOH (99.9%), 2-Propanol, CoO (99.99%) and Co$_3$O$_4$ reference (99.99%) are all from Sigma-Aldrich, Germany. Acetic acid (≥99.0%) for flame spray synthesis was bought from Roth, Switzerland. Nafion (99.9%) solution was originally supplied by Sigma-Aldrich, Germany. It was mixed with

NaOH (99.9%, Sigma-Aldrich, Germany) solution for Na exchange before use in electrochemical measurements.

### Materials synthesis

The Co$_x$Mn$_{1-x}$O$_y$ samples with different nominal Co percentages (x = 1, 0.9, 0.67, 0.5, 0.33, 0.1, and 0) were prepared by the flame spray synthesis (FSS) method reported previously by our group[13,39]. The stoichiometric mixture of Co(NO$_3$)$_2$ and Mn(NO$_3$)$_2$ was first mixed in a solution of ultrapure water and acetic acid (volume ratio is 75:25), with a total metal concentration of 0.6 M. Then the prepared solution was pumped to the nozzle at a flow rate of 20 mL min$^{-1}$ and combusted by the flame. The gas flow rates of oxygen for dispersion, oxygen for combustion, and acetylene for combustion were 35, 17, and 13 L min$^{-1}$, respectively. The powder collected after flame spray pyrolysis was further annealed in a muffle furnace at 500 °C for 4 h.

In addition, the Co-doped CeO$_2$ (Co-CeO$_2$), FeCo$_2$O$_4$, and CoCr$_2$O$_4$ spinel oxides were also synthesized using a similar protocol as above. For the FeCo$_2$O$_4$, the Fe/Co molar ratio is 33:67. For CoCr$_2$O$_4$, the Cr/Co molar ratio is 67:33. For the Co-CeO$_2$, Co(NO$_3$)$_2$ and Ce(NO$_3$)$_3$ were used, with a molar ratio of Co/Ce = 5:95; note that the Co-CeO$_2$ sample is crystalline after FSS and does not need to be annealed at 500 °C for 4 h to obtain the CeO$_2$ phase.

### Structural characterization

XRD patterns were collected in an XRD instrument supplied by Rigaku, Japan. The voltage and current for the Cu anode were set to 40 kV and 160 mA, respectively. The Raman measurement was performed on a LabRAM Series Raman Microscope (Horiba Jobin Yvon) with a He−Ne laser (λ: 632.8 nm, output power: ~20 mW). The XPS spectra were collected on a VG ESCALAB 220iXL spectrometer (Thermo Fisher Scientific), with the focused monochromatized Al Kα radiation (1486.6 eV) as the source (beam size of ~500 μm$^2$). All the spectra were recorded using a pass energy of 20 eV in steps of 50 meV and a dwell time of 50 ms, under the chamber pressure of ~ 2 × 10$^{-9}$ mbar. The spectrometer was calibrated on a clean silver surface by measuring the Ag 3$d_{5/2}$ peak at a binding energy of 368.25 eV with a full width at half-maximum (fwhm) of 0.78 eV.

### Electrochemical characterizations

The electrochemical measurements were carried out using the Biologic VMP-300 software. A conventional three-electrode setup has been

used, with an Au mesh as the counter electrode. Two electrolytes were used, namely 0.1 M KOH and 0.05 M $H_2SO_4$. The electrolytes were saturated with synthetic air (purity 5.6, PanGas AG, Switzerland) prior to measurement. Accordingly, the Hg/HgO (filled with 0.1 M KOH) and Hg/HgSO$_4$ (filled with saturated $K_2SO_4$) were used as the reference electrodes. The reference electrodes were calibrated to the reversible hydrogen electrode (RHE) with the zero intercept ($\Delta V$) of the cyclic voltammetry (CV) of the hydrogen evolution/oxidation reaction in the corresponding electrolyte, as follows:

$$E_{RHE} = E_{measured} - \Delta V - iR \qquad (1)$$

The term "iR" comes from the resistance calibration due to the electrolyte. The corresponding $\Delta V$ for Hg/HgO in 0.1 M KOH and Hg/HgSO$_4$ in 0.05 M $H_2SO_4$ are −0.93 V and −0.716 V, respectively.

To prepare the Thin-Film RDE working electrode[40], the catalysts were first mixed with the solution of water, 2-propanol and Na-exchanged Nafion (volume ratio = 200:50:1) under sonication, to obtain a concentration of 2 g L$^{-1}$. Then 10 µL of the catalyst ink was dropcasted onto a rotating disk electrode (with a glassy carbon disk substrate with 5 mm diameter) to obtain a mass loading of 0.02 mg. The working electrode was dried at ambient conditions.

### Measurement protocol

The measurement of OER activity followed a 19-step protocol. The working electrode is held at open circuit potential for 15 s, then the electrochemical impedance spectroscopy (EIS) is performed at 1 V vs. RHE. In the third step, the catalyst is activated by performing cyclic voltammetry (CV) at the appropriate potential window for 10 cycles. The first 5 CVs are performed at a scan rate of 100 mV s$^{-1}$, the last 5 CVs are performed at the scan rate of 50 mV s$^{-1}$. The next 4th to 17th steps (thus 15 data points) are the chronoamperometric (CA) measurements at different potentials to derive the steady-state polarization curves and Tafel plots. The 18th and 19th steps are again the EIS measurements at 1 V and 1.6 V vs. RHE. The average solution resistance from the three EIS measurements is used for the iR correction.

### Ex situ soft XAS measurement

The ex situ X-ray adsorption spectroscopy measurements were performed in the Phoenix beamline in Swiss Light Source (SLS), Paul Scherrer Institute (PSI), Villigen, Switzerland. The beamline source is an elliptical undulator. The low-energy branch line utilizes the planar grating monochromator and the optics from the X-Treme beamline[41]. The flux of the incoming light is derived from the total electron yield (TEY) signal of an Au TEM grid located upstream of the sample, the electrical current from the sample serves as the TEY signal from the sample. Both signals were measured using a Keithley current amplifier. The sample is measured in a vacuum of ~10$^{-6}$ mbar.

For the general measurement, the catalyst powder was loaded on a conductive carbon to achieve high conductivity on a Cu plate. For the reconstructed CoO, the catalyst ink was dropcasted on the glassy carbon disk and dried, and then the glassy carbon disk was attached on the Cu plate with a conductive tape. All the soft XAS spectra were collected in the total electron yield (TEY) mode.

The data analysis was performed in Athena[42]. For the Co L edge, the parameters for the pre-edge energy range were set to −17.94 to −7.94 eV; the parameters for the normalization range were set to 15 to 72 eV. For the O K edge, the parameters for the pre-edge range were set to −10.46 to −2.80 eV; the parameters for the normalization range were set to 22.15 to 29.54 eV.

### Operando/ex situ hard XAS measurement

The ex situ and operando hard XAS measurements were performed in the SuperXAS beamline at SLS, PSI, Villigen, Switzerland. The ex situ measurements were performed on the pellets made from the powder

of the catalysts. The operando measurements were performed in a home-made spectro-electrochemical cell[43]. The catalyst ink was sprayed on an Au-coated Kapton foil as a working electrode. Carbon black were used when preparing the counter electrode. A special Ag/AgCl electrode was used as the reference electrode. The scan rate for CV measurement was 4 mV s$^{-1}$, while two XAS spectra were collected per second. Every 20 spectra are averaged into 1 during data analysis, to obtain a resolution of 40 mV for the applied potential. The Co K edge position was also extracted using an integral method[6,31] and plotted as a function of applied potential.

## Data availability

The original data for Figs. 1–4 in the main text is available in the Materials Cloud: https://doi.org/10.24435/materialscloud:v8-hq. All other data are available from the corresponding authors upon request. Source data are provided with this paper.

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

## Acknowledgements

The soft X-ray adsorption spectroscopy and hard X-ray adsorption spectroscopy characterizations were carried out in the Phoenix X07MB and SuperXAS beamline, at the SLS, PSI, Villigen, Switzerland. E.F. and J.H. gratefully acknowledge the Swiss National Science Foundation through its PRIMA grant (grant No. PR00P2_193111) and the NCCR MARVEL, a National Centre of Competence in Research, funded by the Swiss National Science Foundation.

## Author contributions

J.H. and E.F. developed the concept. E.F. guided the experiment. J.H. conducted the electrochemical and spectroscopy characterizations and analyzed the data. C.N.B. and T.H. helped to collect the soft X-ray adsorption spectroscopy data. N.S.Y. helped J.H. on materials synthesis. D.B. and M.E.K. helped to perform the XPS characterization. C.W.S. helped to collect the Raman data. E.F. and T.J.S. applied the funding to support the research. All the authors have revised the manuscript.

## Competing interests

The authors declare no competing interests.
