## [Peer Review File · Nature Communications]

REVIEWER COMMENTS

Reviewer #1 (Remarks to the Author):

This manuscript reports the investigation of CoMn-based spinel oxides with tunable Co³⁺/Co²⁺ surface ratio for oxygen evolution reaction. Authors suggested that the catalytic activity is dependent on the presence of low-spin Co³⁺ to facilitate the surface reconstruction and initiate the OER in acidic environment by surface soft X-ray XAFS. However, some points should be modified or considered enough, as listed below. Therefore, I think that this manuscript is not suitable for "Nature Communications". I recommend publishing in another specialized journal.

i) I do not know the novelty of this manuscript enough as "Nature Communications". There are many previous works about cobalt oxide electrocatalysts for oxygen evolution reaction. Especially, Ailong Li reported that the Co₂MnO₄ material resulted in a lifetime of over 2 months at 200 mA cm⁻² at pH 1 (Reference 5: Nat. Catal. 5, 109-118 (2022)). Thus, authors need to exhibit the new aspect of CoMn-based spinel oxide materials compared with previous works. Please clearly exhibit the novelty of this manuscript and refer many previous works concerning cobalt oxides.

ii) Authors suggested that the acidic OER activity of Co-based catalysts are determined by their surface oxidation/spin state. If so, authors should compare the electronic states between surface and bulk by Co L-edge and O K-edge XAFS with total electron yield and fluorescence yield, respectively. Co K-edge XAFS with He conversion electron yield (surface) and fluorescence yield (bulk) is also useful to discuss the electronic state and local structure. Moreover, operando observation may be useful by XAFS.

iii) Did authors check the surface state by XPS analysis? The result of surface XAFS need to be consistent with that of XPS.

iv) How did authors decide the Co spin state in Co L-edge XAFS? Authors should simulate the spin state and XAFS spectra by theoretical method.

v) I could not catch the meaning the current density (A g⁻¹). Please show the current density (A cm⁻²) by geometrical surface area.

vi) In Co L-edge XAFS, the spectra of CoOOH and LiCoO₂ are important as Co³⁺ and Co⁴⁺ materials. Please measure these materials and discuss the electronic state.

Reviewer #2 (Remarks to the Author):

In this manuscript, the authors synthesized a series of CoMn-based spinel oxides for the acidic OER. Combined the sXAS characterization and the electrochemical analysis, this work confirmed that the surface oxidation/spin state is the key factor of the OER activity. Furthermore, the presence of low-spin Co^{III} in Co_xMn_{1-x}O_y catalysts is the active sites for the OER in acidic environment, while the high-spin

CoII is not active. This work is meritorious and interesting as it offers a novel strategy to demonstrate the correlation between OER activity and Co oxidation/spin state of Co-based catalysts for acidic OER. However, more evidence should be provided to confirm such conclusion. Based upon authors' responses, this paper should be evaluated again.

1. The authors claimed that the oxidation of CoII to CoIII is unfavorable for acidic OER, and they performed 10 CV cycles to prove this conclusion. However, only 10 CV cycles to perform the chemical reconstruction to form CoIII is not enough, and the authors should perform more characterizations to demonstrate the formation of surface reconstruction.
2. Co dissolution is inevitable in acidic environment, which has a significant influence on the performance of catalysts. In this work, the authors mentioned the easier dissolution of CoII compared to CoIII. More discussion and characterizations should be provided to support this conclusion.
3. The branching ratios were calculated to confirm the spin states of Co ions in $\text{Co}_x\text{Mn}_{1-x}\text{O}_y$ catalysts in this work. The spin state of metals is closely related to the valence state, so the author should conduct more characterization to study the valence states of the materials.
4. As for the sample with 0.67 (MnCo_2O_4), the work (Nature Catalysis 5, 2022, 109–118) reported a superior activity and durability toward acidic OER and PEMWE, and they also concluded the coexistence of Co^{2+} and Co^{3+} . Such phenomenon is contradictory to this work. The authors anticipated such difference was related to the different surface state due to the different synthesis protocol. Hence, more evidences should be given by this work to confirm this conclusion.
5. The ethanol molecule was employed to confirm that the CoII/CoIII catalysts was beneficial to the OH^* formation, so the authors suggested that the poor OER activity of CoII catalysts in the acidic electrolyte was due to unfavorable deprotonation to form CoIII-OH species on the surface. I really can't understand this conclusion well, please give more discussion.
6. There are some obvious minor errors in the article. The authors should carefully check their article to avoid these minor problems.

Reviewer #3 (Remarks to the Author):

This manuscript unveiled the correlation between Co oxidation/spin state and OER activity, which is a significant advancement in the structure-activity relationship of Cobased catalysts for acidic OER. These findings not only aid in the development of efficient acidic OER catalysts but also enhance our comprehension of the reaction mechanism. I suggest that the manuscript can be published after addressing concerns as follow:

- 1) For the $\text{Co}_x\text{Mn}_{1-x}\text{O}_y$ samples, how about the exact element composition and morphology? It is recommended to conduct physical characterizations such as TEM,

BET, or ICP and include the results in the supplementary information (SI). The changes in composition may lead to variations in particle sizes, surface areas, and conductivity of the oxide particles, which are crucial factors influencing the performance. It is worth noting that normalizing the OER currents by mass may potentially provide misleading information regarding the Tafel slope and intrinsic OER activities of these oxide catalysts.

2) Again, for the $\text{Co}_x\text{Mn}_{1-x}\text{O}_y$ samples, Co and Mn elements exhibit instability in acidic conditions, leading to their dissolution in acidic electrolytes. Among the two, Mn could be particularly prone to this phenomenon. Consequently, the intrinsic performance of the catalyst can be attributed solely to the presence of surface Co species. It is therefore advisable to employ surface analysis techniques like XPS to provide a comprehensive understanding of the surface chemistry and structure.

3) Since the author claim that the correlation does not apply to alkaline and neutral environments, the difference may have originated from the distinct surface compositions resulting from the different dissolution behaviors of Co and Mn in acidic and alkaline conditions.

4) In Figure 3, for comparing the dominant CoII or CoIII features in the Co L edge spectra, commercial Co_2O_3 could be employed as a standard sample of CoIII to be examined and compared by soft XAS.

5) The manuscript investigated the surface states of twelve distinct Co-based catalysts, all of which were in pristine conditions. However, it is crucial to consider surface reconstruction during the pre-catalyst stage, as it greatly influences the OER activity. Furthermore, this study specifically compared the OER performance in acid and alkaline solutions, whereas the CoII or CoIII characteristics may vary significantly in these solutions due to surface reconstruction. Therefore, it is highly recommended to conduct either in-situ characterizations or ex situ characterizations on reacted catalysts.

6) In Page 9, the authors claim that “the formation of OH^* is more favorable than in the acidic environment”. It may be arisen from the different reaction steps during OER process in acid and alkaline electrolyte. It is imperative to provide additional discussions and references to delve into the reaction mechanism extensively.

Response Letter

Dear Reviewers,

We thank all of you for the professional comments/suggestions that help to improve the manuscript. We appreciate your time and effort in re-evaluating our manuscript! We have provided the new supplementary data as suggested and improved the rigor of the discussions. We hope that your concerns have been adequately addressed! The point-by-point responses to each comment/suggestion are shown in the following. The revisions made in the main text and Supplementary information are highlighted in yellow.

Reviewer #1 (Remarks to the Author):

This manuscript reports the investigation of CoMn-based spinel oxides with tunable Co³⁺/Co²⁺ surface ratio for oxygen evolution reaction. Authors suggested that the catalytic activity is dependent on the presence of low-spin Co³⁺ to facilitate the surface reconstruction and initiate the OER in acidic environment by surface soft X-ray XAFS. However, some points should be modified or considered enough, as listed below. Therefore, I think that this manuscript is not suitable for “Nature Communications”. I recommend publishing in another specialized journal.

Response: We thank the Reviewer for providing the insightful comments/suggestions after the first round of evaluation! Before going directly into the details, we would like to share our understanding of this project. Proton Exchange Membrane (PEM) electrolyzers are the ideal devices to store the intermittent renewable electricity into Hydrogen. Developing efficient and cost-effective anode catalyst is challenging. In the short-term, the anode catalyst design would be focused on reducing the use of precious Ir or Ru metal. However, in the long term, finding other alternatives is key to the sustainability and scale-up of this technology. Among the 3d-transition-metal oxides, the Co-based catalysts show promising OER performance in acidic environment. However, we found that the structure-activity relationship in acidic environment is different from that found in other electrolytes, which is essential for the catalyst design but has not yet been discussed in the literature. Therefore, we hope to make our new findings available to the community through Nature Communications, which has a high impact and wide reach with open access privileges. We will address the concerns point by point below. We sincerely thank the Reviewer for the support in improving the manuscript and bringing these interesting results to the readers of Nature Communications.

i) I do not know the novelty of this manuscript enough as “Nature Communications”. There are many previous works about cobalt oxide electrocatalysts for oxygen evolution reaction. Especially, Ailong Li reported that the Co₂MnO₄ material resulted in a lifetime of over 2 months at 200 mA cm⁻² at pH 1 (Reference 5: Nat. Catal. 5, 109-118 (2022).). Thus, authors need to exhibit the new aspect of CoMn-based spinel oxide materials compared with previous works. Please clearly exhibit the novelty of this manuscript and refer many previous works concerning cobalt oxides.

Response: We agree with the Reviewer that there are many studies about Co-based oxides for oxygen evolution reaction. Indeed, Co-based catalysts are very promising compared to other 3d-transition-metal oxides in different electrolytes. Considering only the published articles on Co-based catalysts for acidic OER reported in the Nature family recently, we can mention: CoBa-POM (**Ref.10**, *Nat. Chem.*, 2018, 10, 24-30), CoFePbO_x (**Ref.12**, *Nat. Catal.*, 2019, 2, 457-465), Co₃O₄/CeO₂ (**Ref.6**, *Nat. Commun.*, 2021, 12, 3036), Co₂MnO₄ (**Ref.5**, *Nat. Catal.*, 2022, 5, 109-118), Co₃O₄/C (**Ref.11**, *Nat. Commun.*, 2022, 13, 4341) and so on. Therefore, Co-based catalysts for acidic OER do have attracted increasing attention in the past few years. These important papers have all been cited properly in the Introduction of the manuscript (**Page 1**).

However, OER is a very complex process that usually involves surface reconstruction. The knowledge of such a process on Co-based catalysts is not yet comprehensive. After reviewing the above and other related papers in the literature, we found that there are still some key issues to be addressed for acidic OER:

- i) **Which physicochemical parameters control the surface reconstruction in acidic environment?**
- ii) **Why is the performance of Co-based catalyst much worse in acidic environment than in alkaline environment?**
- iii) **What is the active structure towards OER in acidic environment?**

In our group, we have several ongoing projects that try to tackle these unresolved problems for the Co-based catalysts. Especially in this project, for the first time, we found that the surface reconstruction in a series of CoMn-based catalysts is pH-dependent. This pH-dependence can be attributed to a different structure-activity relationship for Co-based catalysts in acidic environment. The catalysts with low Co^{III}/Co^{II} ratio show poor OER activity in acidic environment, due to the unfavorable formation of active species. By using the sXAS and extending the dataset to another 6 representative samples, we found that the surface state of the catalyst is key to determining the surface reconstruction and OER activity in acidic environment.

According to the Reviewer's suggestion, we have adjusted the discussions in the Introduction on **Page 2**, to make general comparison, point out the problem remains in the field and highlight the novelty of this project.

ii) Authors suggested that the acidic OER activity of Co-based catalysts are determined by their surface oxidation/spin state. If so, authors should compare the electronic states between surface and bulk by Co L-edge and O K-edge XAFS with total electron yield and fluorescence yield, respectively. Co K-edge XAFS with the conversion electron yield (surface) and fluorescence yield (bulk) is also useful to discuss the electronic state and local structure. Moreover, operando observation may be useful by XAFS.

Response: We thank the Reviewer for these professional suggestions! We agree with the Reviewer that the spectra collected with total electron yield and fluorescence yield reflect the information at different depth from surface, which would be ~ 5 nm and 100 nm respectively. The fluorescence yield spectra are more bulk-sensitive. It is very unlucky that the Swiss light source (SLS) has been shut down for the upgrade, therefore it is not convenient for us to collect the new data on the fluorescence mode in the short term. In the following, we will use the XAS data we have collected before the shutdown to address the concerns as most as possible.

First, we have added the Co K edge spectra of 4 representative Co-based spinel oxides as the new **Supplementary Figure 11** (also shown below). The Co K edge position (E_{edge}) was extracted using an integral method (**Ref. 6**, *Nat. Commun.* **12**, 3036 (2021); **Ref. 31**, *Anal. Bioanal. Chem.* **376**, 562-583 (2003)), to make comparison with the $I(\text{Co}^{\text{III}})/I(\text{Co}^{\text{II}})$ at the Co L₃ edge spectra (**Supplementary Figure 11c**). The commercial Co₃O₄ (particle size in micron) is a good reference to study the Co oxidation state and local structure. The Co oxidation states of Co_xMn_{1-x}O_y ($x = 1$) and Co_xMn_{1-x}O_y ($x = 0.67$) catalysts are higher and lower than that of commercial Co₃O₄, respectively. All the three catalysts show three typical peaks attributed to spinel structure in the Fourier transform EXAFS spectra, indicating the Co atoms are in both octahedral and tetrahedral coordination (**Supplementary Figure 11b**). The CoCr₂O₄ catalyst is also a good reference for 2+; besides, the Co atoms are in the tetrahedral coordination of spinel structure, which is supported by no obvious peak attributed to the octahedral coordination from EXAFS spectra (**Supplementary Figure 11b**). Generally, higher E_{edge} (i.e., higher Co oxidation state) is associated with higher the $I(\text{Co}^{\text{III}})/I(\text{Co}^{\text{II}})$ at the Co L edge spectra. We have added the related discussion on **Page 5-6**.

Supplementary Figure 11. Hard XAS characterization of Co K edge in four representative Co-based spinel oxides. (a) The XANES spectra, and (b) the corresponding Fourier transform EXAFS

spectra. (c) The extracted Co K edge position was compared with the $I(\text{Co}^{\text{III}})/I(\text{Co}^{\text{II}})$ in Co L edge spectra.

Second, according to the Reviewer's suggestion, we have added the operando hXAS characterization on commercial $\text{Co}(\text{OH})_2$ as the new **Supplementary Figure 19**, to provide the insights on the pH-dependent Co oxidation behavior during OER. The operando experiments were carried out at the SuperXAS beamline, of the Swiss Light Source of Paul Scherrer Institute before the synchrotron shutdown. The experimental details have been added in Methods (**Page 12**). The $\text{Co}(\text{OH})_2$ is clearly active in 0.1 M KOH, but not active in 0.05 M H_2SO_4 , as revealed by the CV profiles (**Supplementary Figure 19a**), which were measured during operando characterization in a spectro-electrochemical cell (**Ref. 43, *J. Electrochem. Soc.* 163, H906-H912 (2016)**). The scan rate for CV measurement is 4 mV s^{-1} , while two XAS spectra per second were collected. Every 20 spectra were averaged into 1 during data analysis, to have a resolution of 40 mV for the applied potential (**Supplementary Figure 19b-c**). The change of Co K edge position (ΔE_{edge}) was also extracted and plotted as a function of applied potential. The hXAS is bulk sensitive and only the surface Co are oxidized during the OER. The changes in Co K edge are small, but the trend is clear different in the alkaline and acidic electrolytes. Obviously, the edge position is shifted to higher energy, when $\text{Co}(\text{OH})_2$ is active in alkaline environment (**red curve in Supplementary Figure 19d**); in contrast, there is no obvious shift at the Co K edge when $\text{Co}(\text{OH})_2$ is not active in acidic environment (**blue curve in Supplementary Figure 19d**), manifesting the Co is difficult to be oxidized to undergo surface reconstruction in acidic environment. We have added the related discussions on **Page 9**.

Supplementary Figure 19. Operando hard XAS characterization of Co K edge in the commercial $\text{Co}(\text{OH})_2$ catalyst to reveal the pH-dependent Co oxidation behavior during OER. (a) The CV profiles collected in a spectro-electrochemical cell, with a scan rate of 4 mV s^{-1} . The operando

XANES spectra of Co K edge collected in (b) 0.1 M KOH and (c) 0.05 M H₂SO₄, respectively. (d) The change of the Co K edge (ΔE_{edge}) is plotted as a function of the applied potential.

iii) Did authors check the surface state by XPS analysis? The result of surface XAFS need to be consistent with that of XPS.

Response: According to the Reviewer's suggestion, we have performed the XPS characterization on the CoMn-based catalysts. The results of Co 2p and O 1s XPS spectra are added as **Supplementary Figure 7**, as shown below. The main peak of Co 2p_{3/2} signal can be deconvoluted into Co^{III} and Co^{II} at ~779.6 to 781.4 eV, respectively. The Co^{III}/Co^{II} ratio on catalyst's surface can be correlated to the area ratio [A(Co^{III})/A(Co^{II})] of the fitting peaks. It shows a similar trend as that of I(Co^{III})/I(Co^{II}) at the Co L₃ edge spectra (**Supplementary Figure 8**). Therefore, the result of soft XAS is consistent with that of XPS.

Supplementary Figure 7. XPS characterization of the Co_xMn_{1-x}O_y samples on the (a) Co 2p and (b) O 1s spectra.

Furthermore, The O 1s spectra can be deconvoluted into three oxygen species: O1 is the O in the lattice, O2 is the undercoordinated oxygen defect, and O3 represents the OH-/H₂O adsorbed on catalyst. Generally, the peak intensity of lattice O decreases significantly when x < 0.9 (O1 in **Supplementary Figure 7b**), which can be also correlated to the decrease in peak intensity related to Co^{III}-O hybridization in Co L₃ edge (Fig. 2b). We thank the reviewer for the nice suggestion to improve the reliability of the manuscript. We have added the new supplementary figures in the Supplementary information, and the related discussions on **Page 6** of the main text.

Supplementary Figure 8. Comparison of the area ratio of $A(\text{Co}^{\text{III}})/A(\text{Co}^{\text{II}})$ at the Co 2p (XPS) spectra and intensity ratio of $I(\text{Co}^{\text{III}})/I(\text{Co}^{\text{II}})$ at the Co L edge spectra (sXAS), respectively.

iv) How did authors decide the Co spin state in Co L-edge XAFS? Authors should simulate the spin state and XAFS spectra by theoretical method.

Response: Thanks for the comment! The Co spin state is decided by the branching ratio of $I(L_3)/[I(L_3) + I(L_2)]$, in which $I(L_3)$ and $I(L_2)$ are the strongest peak intensity of Co L_3 and L_2 spectra, respectively. In the literature, the $\text{Co}(\text{OH})_2$ is a standard sample of Co atoms in the high spin state, with a branching ratio of ~ 0.7 ; the calculated branching ratio for the commercial $\text{Co}(\text{OH})_2$ in this manuscript is ~ 0.74 , confirming that the Co atoms are in high spin state. In comparison, the branching ratio for Co atom in low spin state is ~ 0.6 . We have cited the related papers to support the discussions on the Co spin state. For the discussion on the standard samples, please see the **Response to the number (vi) comment in the following.**

Furthermore, we note that the data analysis would slightly influence the calculated branching ratio. To prove the rigor of our data processing, all the Co L edge spectra reported in this manuscript have applied the same parameters for the normalization in Athena software. The parameters are listed in Method (**Page 12**). In addition, the $I(L_3)$, $I(L_2)$ and $I(L_3)/[I(L_3) + I(L_2)]$ are also provided in **Supplementary Table 1 and 4**. We thank the Reviewer for the nice suggestion to have theoretical simulation. The simulated spectra for Co L edge with Co of high/low spin state have already been well explored by Merz et al. (**Ref. 32**, Physical Review B 82, 174416, 2010) and other groups. We have cited the paper and add the discussions on **Page 6-7**, as in the following:

“Among the investigated samples, commercial $\text{Co}(\text{OH})_2$, Co-CeO₂ (Supplementary Figure 12) and CoCr₂O₄ (Supplementary Figure 13) prepared by FSS show only the typical features of Co^{II}, similar to the simulated spectra of Co^{II} high spin state³².”

v) I could not catch the meaning the current density ($A g^{-1}$). Please show the current density ($A cm^{-2}$) by geometrical surface area.

Response: Thanks for the Comment! We have provided another dataset similar to Fig.1a-d but normalized by the geometric surface area in the Supplementary information, as shown in **Supplementary Figure 3** (also shown in the following). For each measurement, 0.02 mg of the catalyst was drop-cast on a glassy carbon disk with the surface area of $0.19625 cm^{-2}$, to form a thin catalyst layer (see Method for more details). The observed trend for polarization curve is not affected by the different normalization methods. Moreover, the Tafel slope, which used as the major parameter for comparison in this project, would not be affected by the normalization method, since

$$0.02 mg \times J(A g^{-1}) = 0.19625 cm^{-2} \times J(mA cm^{-2})$$

Then

$$\text{Log } [J(A g^{-1})] = \text{Log}[J(mA cm^{-2})] + \text{Log}(9.8125)$$

$$\frac{\partial \text{Log } [J(A g^{-1})]}{\partial V} = \frac{\partial \text{Log } [J(mA cm^{-2})]}{\partial V}$$

$$\text{Tafel slope} = \frac{\partial V}{\partial \text{Log } [J(A g^{-1})]} = \frac{\partial V}{\partial \text{Log } [J(mA cm^{-2})]}$$

where V is the potential. Therefore, the different normalization methods only cause the Tafel plot shift parallel along the x axis but would not change the Tafel slope from the fit.

Supplementary Figure 3. Comparison of the OER activity of $\text{Co}_x\text{Mn}_{1-x}\text{O}_y$ ($x = 1, 0.9, 0.67, 0.5, 0.33, 0.1$ and 0) in the (a) alkaline and (b) acidic environments. (c-d) The corresponding Tafel plots. The current is normalized by the geometric area of electrode, in comparison to the results normalized by the catalyst mass loading in Fig. 1.

vi) In Co L-edge XAFS, the spectra of CoOOH and LiCoO_2 are important as Co^{3+} and Co^{4+} materials. Please measure these materials and discuss the electronic state.

Response: We would like to thank the Reviewer for the nice suggestion! The CoOOH and LiCoO_2 (Li-deficient) are good reference samples for the Co^{3+} and Co^{4+} , respectively. We do not have the suggested LiCoO_2 sample at hand to have standard spectra for Co^{4+} at this state. Luckily, we have CoOOH as the reference sample in the group, therefore we use it as the standard Co^{3+} sample. The Co L edge spectra and the corresponding branching ratios are added as new **Supplementary Fig. 6** (also shown below). The Co^{2+} in $\text{Co}(\text{OH})_2$ is suggested to be in high spin state (Ref. 26, ACS Appl. Nano Mater. 2022, 5, 18680–18690), while the Co^{3+} in the pristine CoOOH is in low spin state (Ref. 27, J. Mater. Chem. A, 2021, 9, 17749). Then the calculated branching ratios for $\text{Co}(\text{OH})_2$ and CoOOH are ~ 0.74 and ~ 0.61 , respectively, well matching the value reported in literature. The related discussion has been added on Page 5, as in the following:

“In literature, Co atoms in $\text{Co}(\text{OH})_2$ and CoOOH are suggested to be in high-spin and low-spin configurations^{26,27}, respectively, with calculated branching ratios of 0.74 and 0.61 (Supplementary Figure 6), proving that the branching ratio is a reliable descriptor of Co spin state.”

Supplementary Figure 6. Standard samples with Co atoms in the high and low spin states. (a) The Co L edge spectra of commercial Co(OH)₂ and CoOOH reference. (b) The calculated branching ratios.

We want to note that CoOOH is also active for OER in acidic environment (Figure R1 in the following). However, the electrochemical data on CoOOH is related to another manuscript that we plan to submit soon, therefore, we would like to not include the discussion on electrochemical performance of CoOOH in the present manuscript. We hope to have your understanding, thank you!

Figure R1. The CV of CoOOH in acidic environment, in comparison to that of Co(OH)₂.

Reviewer #2 (Remarks to the Author):

In this manuscript, the authors synthesized a series of CoMn-based spinel oxides for the acidic OER. Combined the sXAS characterization and the electrochemical analysis, this work confirmed that the surface oxidation/spin state is the key factor of the OER activity. Furthermore, the presence of low-spin Co^{III} in Co_xMn_{1-x}O_y catalysts is the active sites for the OER in acidic environment, while the high-spin Co^{II} is not active. This work is meritorious and interesting as it offers a novel strategy to demonstrate the correlation between OER activity and Co oxidation/spin state of Co-based catalysts for acidic OER. However, more evidence should be provided to confirm such conclusion. Based upon authors' responses, this paper should be evaluated again.

Response: We sincerely thank the Reviewer for approving the manuscript! With the help of the Reviewer, we understand that there are many aspects that could be improved in the manuscript, especially as it is prepared in a short communication format. We are grateful for the opportunity to revise the manuscript! We hope that with the new supplementary data below, the reliability of the manuscript has been improved!

1. The authors claimed that the oxidation of Co^{II} to Co^{III} is unfavorable for acidic OER, and they performed 10 CV cycles to prove this conclusion. However, only 10 CV cycles to perform the chemical reconstruction to form Co^{III} is not enough, and the authors should perform more characterizations to demonstrate the formation of surface reconstruction.

Response: We would like to thank the Reviewer for the good suggestion! It is well accepted in literature that surface reconstruction happens during OER. The Co^{II} in tetrahedral coordination is major responsible for the formation of CoOOH species in alkaline environment (**Ref.17**, *J. Am. Chem. Soc.* 2016, 138, 36–39). According to the Reviewer's suggestion, we have performed Raman characterization on the CoCr₂O₄ catalyst under four different conditions: on the pristine nano powder, on the catalyst after ink preparation, and on the catalyst after OER in 0.1 M and 0.05 M H₂SO₄, respectively. The representative peaks of the CoCr₂O₄ spinel structure are maintained after OER in both acidic or alkaline environments (**Supplementary Figure 18**, as demonstrated below); the peak position is shifted to higher wavenumber, possibly due to strains that are generated at the surface due to the reconstruction or metal dissolution. It is a pity that the representative peaks for the Co (oxy)hydroxide are in range of ~400 to 600 cm⁻² (*J. Am. Chem. Soc.* 2020, 142, 11901–11914), which is overlapping with those from CoCr₂O₄, making it difficult to distinguish whether new CoO_xH_y species are present on the surface. However, we can still extract two important findings from these results: i) the surface reconstruction in alkaline environment is limited, possibly only a thin layer of CoO_xH_y with poor crystallinity is formed; ii) the catalyst is still maintained after OER in acidic electrolyte, therefore the poor OER activity is not due to dramatic catalyst dissolution. Based on these new findings, we have reorganized the related discussion in the main text on **Page 8**:

“Raman characterization reveals that the CoCr₂O₄ spinel structure is well-maintained after OER in both alkaline and acidic electrolyte (Supplementary Figure 18), indicating i) the surface

reconstruction in alkaline environment is limited; ii) the poor OER activity in acidic environment is not entirely due to catalyst dissolution.”

Supplementary Figure 18. Raman characterization on the pristine CoCr_2O_4 nano-powder and its counterparts after ink preparation, and after OER in 0.1 M KOH and 0.05 M H_2SO_4 , respectively.

2. Co dissolution is inevitable in acidic environment, which has a significant influence on the performance of catalysts. In this work, the authors mentioned the easier dissolution of Co^{II} compared to Co^{III} . More discussion and characterizations should be provided to support this conclusion.

Response: We thanks the Reviewer for bring this statement to our attention! In the first version of the manuscript, we tried to understand that the poor OER activity of Co^{II} catalysts compared to $\text{Co}^{\text{II}}/\text{Co}^{\text{III}}$ catalyst could be partially due to the easier Co dissolution in the Co^{II} catalysts. However, we notice it is not a rigorous discussion since the Co dissolution is dependent on both the structure and the operating conditions (e.g., applied potential, electrolytes and so on). It is difficult to make a fare comparison by doing more experiments. To avoid the misleading to the readers, we have adjusted the discussion on **Page 8**, as the following:

“The Co-based catalysts are more stable in alkaline environment; in comparison, the Co dissolution is inevitable in acidic environment, and it is dependent on the structure properties and the operating conditions (e.g., applied potential, electrolytes and so on)^{5,6}. We would like to point out that the poor OER activity of Co^{II} catalysts is not due to a significant sample dissolution during measurement.”

3. The branching ratios were calculated to confirm the spin states of Co ions in $\text{Co}_x\text{Mn}_{1-x}\text{O}_y$ catalysts in this work. The spin state of metals is closely related to the valence state, so the author should conduct more characterization to study the valence states of the materials.

Response: We would like to thank the reviewer for the great suggestions. We have performed the XPS characterization to look at the Co 2p spectra (**Supplementary Figure 7**). The deconvolution results for Co^{III} to Co^{II} ratio show a trend similar to that observed at the Co L_3 edge spectra (**Supplementary Figure 8**). These new results will be compared with the soft XAS results to have a more insightful discussion on the electronic structures of the catalysts. The related discussions are added on **Page 6**.

Supplementary Figure 7. XPS characterization of the $\text{Co}_x\text{Mn}_{1-x}\text{O}_y$ samples on (a) Co 2p and (b) O 1s spectra.

In addition, the Co K edge spectra of 4 representative spinel oxides have been added as a new **Supplementary Figure 11**. The commercial Co_3O_4 (particle size in micron) and CoCr_2O_4 catalyst are the good reference samples for the Co oxidation state of 2.67+ and 2+, respectively. The Co K edge position (E_{edge}) can be extracted to make a qualitative comparison. It shows that higher E_{edge} (i.e., higher Co oxidation state) is associated with the higher $I(\text{Co}^{\text{III}})/I(\text{Co}^{\text{II}})$ at the Co L edge spectra. The related discussions on the Co K edge are added on **Page 6** of the main text.

Supplementary Figure 8. Comparison of the area ratio of A(Co^{III})/A(Co^{II}) at the Co 2p (XPS) spectra and intensity ratio of I(Co^{III})/I(Co^{II}) at the Co L edge spectra (sXAS), respectively.

Supplementary Figure 11. Hard XAS characterization of Co K edge in four representative Co-based spinel oxides. (a) The XANES spectra, and (b) the corresponding Fourier transform EXAFS spectra. (c) The extracted Co K edge position (E_{edge}) was compared with the I(Co^{III})/I(Co^{II}) in Co L edge spectra.

4. As for the sample with 0.67 (MnCo_2O_4), the work (Nature Catalysis 5, 2022, 109–118) reported a superior activity and durability toward acidic OER and PEMWE, and they also concluded the coexistence of Co^{2+} and Co^{3+} . Such phenomenon is contradictory to this work. The authors

anticipated such difference was related to the different surface state due to the different synthesis protocol. Hence, more evidences should be given by this work to confirm this conclusion.

Response: We thank the reviewer for bring this issue to our attention. We have used a different protocol (flame spray pyrolysis + annealing) compared to the one reported previously for MnCo_2O_4 . To get more details on the as-prepared $\text{Co}_x\text{Mn}_{1-x}\text{O}_y$ ($x = 0.67$) in this manuscript, we first provided the Co K edge spectra (**Supplementary Figure 11**, as shown above) to study the bulk Co oxidation state, to show it has lower Co oxidation state compared to the commercial Co_3O_4 . In addition, we have studied the surface Co/Mn atomic ratio by XPS analysis; the measured Co atomic percentage (55.5%) is lower than the nominal value (66.7%). More importantly, the $I(\text{Co}^{\text{III}})/I(\text{Co}^{\text{II}})$ of $\text{Co}_x\text{Mn}_{1-x}\text{O}_y$ of $x = 0.67$ is only ~ 1.19 , much lower that of commercial Co_3O_4 (~ 2.24), suggesting the surface Co^{III} is much less than that of Co_3O_4 . To clarify the difference and avoid the misleading, we have added the following discussion on **Page 6**:

“The lower Co oxidation state of $\text{Co}_x\text{Mn}_{1-x}\text{O}_y$ with $x = 0.67$ compared to Co_3O_4 , and the lower Co atomic percentage (55.5%, Supplementary Figure 9) on the surface could account for the lower OER activity of this material compared to CoMn_2O_4 catalyst in literature⁵. In particular, the $I(\text{Co}^{\text{III}})/I(\text{Co}^{\text{II}})$ of $\text{Co}_x\text{Mn}_{1-x}\text{O}_y$ with $x = 0.67$ is only ~ 1.19 , much lower than that of commercial Co_3O_4 (~ 2.24).”

Supplementary Figure 9. The comparison of nominal Co atomic percentage used for synthesis and the Co atomic percentage measured by XPS. The Co and Mn atomic percentages measured by XPS are also summarized in Supplementary table 5.

5. The ethanol molecule was employed to confirm that the $\text{Co}^{\text{II}}/\text{Co}^{\text{III}}$ catalysts was beneficial to the OH^* formation, so the authors suggested that the poor OER activity of Co^{II} catalysts in the acidic electrolyte was due to unfavorable deprotonation to form $\text{Co}^{\text{III}}\text{-OH}$ species on the surface. I really can't understand this conclusion well, please give more discussion.

Response: We would like to thank the Reviewer for bring this point to our attention! The use of alcohol molecules as chemical probe is proposed by Prof. Bin Liu and his colleagues (**Ref. 38**, *Joule* **3**, 1498-1509 (2019)). It is suggested that the OH^* (the first intermediate for OER) is electrophilic and could partially oxidize alcohol to aldehyde or acid. Usually, the alcohol oxidation will happen before OER to have an earlier onset in the polarization curve, if OH^* are already generated on the surface of the catalyst. In contrast, if the OER is limited by the formation of OH^* , then there would be no obvious difference in the polarization curves with/without alcohol in the electrolytes. It is also suggested the binding energy of the catalyst to OH^* is very weak under latter situation. This method is very usefully to check whether there are OH^* generated before the onset of OER. To better explain the merits of this method, we have reformed the related discussion on **Page 9-10** as the following:

“ OH^* is the first intermediate of the OER; it is electrophilic and can oxidize ethanol into aldehyde or acid³⁸. Therefore, the onset of the OER polarization curve will be earlier in the electrolyte with ethanol if there are already OH^* generated on the catalyst’s surface; otherwise, the catalyst with weak OH^* binding strength is limited by the OH^* formation and shows no obvious difference between polarization curves after adding ethanol into the electrolyte.”

6. There are some obvious minor errors in the article. The authors should carefully check their article to avoid these minor problems.

Response: We thank the Reviewer for bringing this problem to our attention! We are sorry for the errors in the manuscript, we understand it really affects the reading experience. In this revised version, we have done our best to check the language and all the figures carefully, hoping to present the data rigorously and accurately, and to make the reader feel comfortable while reading. We appreciate your support in re-evaluating our manuscript!

Reviewer #3 (Remarks to the Author):

This manuscript unveiled the correlation between Co oxidation/spin state and OER activity, which is a significant advancement in the structure-activity relationship of Co-based catalysts for acidic OER. These findings not only aid in the development of efficient acidic OER catalysts but also enhance our comprehension of the reaction mechanism. I suggest that the manuscript can be published after addressing concerns as follow:

Response: We thank the Reviewer for his/her time and efforts on evaluating our manuscript! Thanks to the nice suggestions, we have conducted new experiments and significantly improved the solidity of our manuscript!

1) For the $\text{Co}_x\text{Mn}_{1-x}\text{O}_y$ samples, how about the exact element composition and morphology? It is recommended to conduct physical characterizations such as TEM, BET, or ICP and include the results in the supplementary information (SI). The changes in composition may lead to variations in particle sizes, surface areas, and conductivity of the oxide particles, which are crucial factors influencing the performance. It is worth noting that normalizing the OER currents by mass may potentially provide misleading information regarding the Tafel slope and intrinsic OER activities of these oxide catalysts.

Response: We thank the Reviewer for the insightful comments! The major concern of the Reviewer is the changes in element composition and morphology might influence the OER performance.

Supplementary Figure 9. The comparison of nominal Co atomic percentage used for synthesis and the Co atomic percentage measured by XPS. The Co and Mn atomic percentages measured by XPS are also summarized in Supplementary table 5.

First, we have conducted the XPS characterization to calculate the Co/Mn ratio on the surface. We notice there are mismatches between the Co atomic percentage measured by XPS compared to the nominal Co atomic percentage used for synthesis (**Supplementary Figure 9**). Therefore, we have adjusted the statement in the main text to avoid misleading to the readers.

Secondly, we agree with Reviewer that the morphology (e.g., particle sizes and surface areas) could affect the catalytic performance. The flame spray synthesis used in this project is a well-developed technique to produce nano particles in our group. The direct measurement on morphology by TEM or SEM is difficult to provide a quantitative comparison among the samples. To address this concern, we have performed the characterization of double layer capacitance (C_{dl}), which is a good indicator of the electrochemical surface area, on all the $\text{Co}_x\text{Mn}_{1-x}\text{O}_y$ samples (**Supplementary Figure 4**). We notice that adding the Mn into the sample significantly decreases the C_{dl} . The C_{dl} for the sample without Mn ($x = 1$) is $\sim 2.48 \text{ F g}^{-1}$; it is immediately decreased to 0.5 F g^{-1} for the sample of $x = 0.9$ (**Supplementary Figure 5**). However, both samples ($x = 1$ and 0.9) show similar OER activity in acidic electrolyte (Figure 1). Therefore, varying the Co/Mn ratio during synthesis changes the C_{dl} of the catalysts; however, this change is not the main factor to determine if the catalyst is active or not in acidic electrolyte. The sXAS and XPS characterizations confirm the surface electronic properties is more crucial to determining the electrocatalytic OER performance.

Supplementary Figure 4. (a-g) The CV profiles collected at difference scan rate (200, 150, 100, 75, 50 and 25 mV s^{-1}) for the $\text{Co}_x\text{Mn}_{1-x}\text{O}_y$ samples ($x = 1, 0.9, 0.67, 0.5, 0.33, 0.1$ and 0). The potential window is between 1.30 to 1.40 V vs. RHE.

Supplementary Figure 5. (a) The current density difference (Δj) at 1.35 V vs. RHE extracted from the CVs in Supplementary Figure 4, is plotted as a function of the scan rates. (b) The extracted C_{dl} is plotted as a function of x in the $\text{Co}_x\text{Mn}_{1-x}\text{O}_y$ catalysts. The error bars are from fitting.

Finally, we have used a quite low mass loading for the electrochemical measurement: 0.02 mg catalyst on a RDE disk with a surface area of 0.19625 cm². The normalization method will not change the trend we observed among the samples. In addition, we have provided a dataset that normalized by the geometric surface, not by the mass, as the **Supplementary Figure 3**. The Tafel slope is the same as those extracted in Figure 1.

Supplementary Figure 3. Comparison of the OER activity of $\text{Co}_x\text{Mn}_{1-x}\text{O}_y$ ($x = 1, 0.9, 0.67, 0.5, 0.33, 0.1$ and 0) in (a) alkaline and (b) acidic environments. (c-d) The corresponding Tafel plots.

The current is normalized by the geometric area of electrode, in comparison to the results normalized by the catalyst mass loading in Fig. 1.

2) Again, for the $\text{Co}_x\text{Mn}_{1-x}\text{O}_y$ samples, Co and Mn elements exhibit instability in acidic conditions, leading to their dissolution in acidic electrolytes. Among the two, Mn could be particularly prone to this phenomenon. Consequently, the intrinsic performance of the catalyst can be attributed solely to the presence of surface Co species. It is therefore advisable to employ surface analysis techniques like XPS to provide a comprehensive understanding of the surface chemistry and structure.

Response: We would like to thank the Reviewer for the nice suggestion! We have first performed the surface-sensitive XPS characterization to take a look at the Co 2p and O 1s XPS spectra (**Supplementary Figure 7**). The deconvolution results for the Co^{III} to Co^{II} ratio show a trend similar to that observed at the Co L_3 edge spectra (**Supplementary Figure 8**). The decrease of lattice oxygen content in O 1s spectra (O1 in **Supplementary Figure 7b**) is also comparable the decrease of peak intensity related to Co^{III} -O hybridization in O K edge spectra (**Fig. 2b**). These new correlations have been discussed in the main text on **Page 5-6**.

Supplementary Figure 7. XPS characterization of the $\text{Co}_x\text{Mn}_{1-x}\text{O}_y$ samples on (a) Co 2p and (b) O 1s spectra.

Supplementary Figure 8. Comparison of the area ratio of A(Co^{III})/A(Co^{II}) Co 2p (XPS) spectra and intensity ratio of I(Co^{III})/I(Co^{II}) in Co L edge spectra (sXAS), respectively.

3) Since the author claim that the correlation does not apply to alkaline and neutral environments, the difference may have originated from the distinct surface compositions resulting from the different dissolution behaviors of Co and Mn in acidic and alkaline conditions.

Response: We thank the reviewer for bringing a new perspective to understand the pH-dependence of OER activity. We agree with the Reviewer that the dissolution behaviors of Co and Mn are dependent on the pH of the electrolyte. According to the Pourbaix diagrams, Mn is not stable at high potential in alkaline environment; in acidic environment, both Co and Mn are not very stable. However, we want to note that Co/Mn dissolution in acidic environment is inevitable, no matter the catalyst is active or not. Therefore, it is difficult to have the clear correlations between the Co/Mn dissolution and the OER performance. We agree with the Reviewer that the surface compositions could be different in acidic and alkaline conditions, which is affected by the surface reconstruction. At last, searching for the parameters that control the surface reconstruction is key to understanding the pH-dependent OER activity. To emphasize the connections between electrolytes, structure properties and surface reconstruction, we have added on the following discussions on **Page 3-4** of the main text:

“The discrepancy in the Tafel slopes in acidic vs. alkaline electrolytes supports the initial hypothesis that the OER mechanism depends on the electrocatalyst/electrolyte interfacial characteristics after surface reconstruction has occurred. The reconstruction process is controlled by both the physico-chemical properties of the pristine electrocatalyst and its interactions with electrolyte.”

4) In Figure 3, for comparing the dominant Co^{II} or Co^{III} features in the Co L edge spectra, commercial Co₂O₃ could be employed as a standard sample of Co^{III} to be examined and compared by soft XAS.

Response: We would like to thank the Reviewer for the nice suggestion! We do not have the suggested Co_2O_3 XAS characterization at hand, but we have the CoOOH as the reference sample in the group, therefore we used it as the standard Co^{III} sample. The Co L edge spectra and the corresponding branching ratios are added as new **Supplementary Figure 6** (also shown below). The calculated branching ratio for CoOOH is ~ 0.61 , well matching the value reported in the literature and confirming the Co atoms of CoOOH are in the low spin state. We proved that CoOOH was also active for OER in acidic environment (Figure R1 in the following). However, the electrochemical data on CoOOH is related to another manuscript that we plan to submit soon, therefore, we would like to not include the discussion on electrochemical performance of CoOOH in the present manuscript. We hope to have your understanding!

Supplementary Figure 6. Standard samples with Co atoms in the high and low spin states. (a) The Co L edge spectra of commercial $\text{Co}(\text{OH})_2$ and CoOOH reference. (b) The calculated branching ratios.

Figure R1. The CV of CoOOH in acidic environment, in comparison to that of the commercial $\text{Co}(\text{OH})_2$.

5) The manuscript investigated the surface states of twelve distinct Co-based catalysts, all of which were in pristine conditions. However, it is crucial to consider surface reconstruction during

the pre-catalyst stage, as it greatly influences the OER activity. Furthermore, this study specifically compared the OER performance in acid and alkaline solutions, whereas the Co^{II} or Co^{III} characteristics may vary significantly in these solutions due to surface reconstruction. Therefore, it is highly recommended to conduct either in-situ characterizations or ex situ characterizations on reacted catalysts.

Response: We agree with the Reviewer that there could be chemical reconstruction in some of the catalyst during ink preparation. The CoO is a very typical catalyst that undergoes chemical surface reconstruction, we have cited the related literature and discussed this phenomenon on Page 6.

In addition, it is well recognized that the surface of Co₃O₄ (with both Co^{II} and Co^{III}) would involve into CoOOH during OER in acidic environment in literature (Ref. 6, Nat Commun 12, 3036 (2021); Ref. 8, J. Am. Chem. Soc. 2023, 145, 14, 7829–7836). However, the situation is not clear for the sample that only has Co^{II}. Therefore, we have performed the Raman characterization on CoCr₂O₄ spinel oxide under four different conditions: on the pristine nano powder, on the catalyst after ink preparation, and on the catalyst after OER in 0.1 M and 0.05 M H₂SO₄, respectively. We want to note that the peak position is shifted to higher wavenumber after ink preparation and after OER in both acidic and alkaline environment, possibly due the strains that are generated at the surface due to the reconstruction or metal dissolution (Supplementary Figure 18). It is a pity that the representative peaks for the Co (oxy)hydroxide are in range of ~400 to 600 cm⁻² (J. Am. Chem. Soc. 2020, 142, 11901–11914), which is overlapping with those from CoCr₂O₄, making it difficult to distinguish if the new CoO_xH_y species are present on the surface.

Supplementary Figure 18. Raman characterization on the pristine CoCr₂O₄ nano-powder and its counterparts after ink preparation, after OER in 0.1 M KOH and 0.05 M H₂SO₄, respectively.

Furthermore, we have added the results of operando hXAS characterization on commercial Co(OH)₂ as the new Supplementary Figure 19, to provide insights on the pH-dependent Co oxidation behavior during OER. The operando experiments were carried out at the

SuperXAS beamline, Swiss Light Source of Paul Scherrer Institute before its shutdown. The experimental details have been added in Methods. The Co(OH)_2 is clearly active in 0.1 M KOH, but not active in 0.05 M H_2SO_4 , as revealed by the CV profiles measured during operando characterization in a spectro-electrochemical flow cell (**Supplementary Figure 19a**). The scan rate for CV measurement is 4 mV s^{-1} , while two XAS spectra were collected per second. Every 20 spectra are averaged into 1 during data analysis, to have resolution of 40 mV for the applied potential (**Supplementary Figure 19b-c**). The change of Co K edge position (ΔE_{edge}) was also extracted and plotted as a function of applied potential. Obviously, the edge position is shift to higher energy, when Co(OH)_2 is active in alkaline environment (**red curve in Supplementary Figure 19d**); in contrast, there is no obvious shift at the Co K edge when Co(OH)_2 is not active in acidic environment (**blue curve in Supplementary Figure 19d**), manifesting the Co is difficult to be oxidized to undergo surface reconstruction in acidic environment. We have added the related discussion on **Page 9**.

Supplementary Figure 19. Operando hard XAS characterization at the Co K edge for commercial Co(OH)_2 catalyst to reveal the pH-dependent Co oxidation behavior during OER. (a) The CV profiles collected in a spectro-electrochemical flow cell, with a scan rate of 4 mV s^{-1} . The operando XANES spectra of Co K edge collected in (b) 0.1 M KOH and (c) 0.05 M H_2SO_4 , respectively. (d) The change of the Co K edge (ΔE_{edge}) is plotted as a function of applied potential.

6) On Page 9, the authors claim that “the formation of OH^* is more favorable than in the acidic environment”. It may be arisen from the different reaction steps during OER process in acid and

alkaline electrolyte. It is imperative to provide additional discussions and references to delve into the reaction mechanism extensively.

Response: We thank the Reviewer for the nice suggestion! We agree with the Reviewer that the rate determining step could be different and Tafel slope is a good indicator. We have cited the related literature and added the following discussion on **Page 10**:

“Therefore, the OH* formation is not likely to be the RDS for all the Co^{II} and Co^{II}/Co^{III} catalysts in an alkaline environment. In an acidic environment, the RDS for Co^{II}/Co^{III} catalysts (Tafel slope of ~ 90 mV dec⁻¹) could be the O-O bond formation⁵; the Co^{II} catalysts with very poor OER activity in the acidic electrolyte (Tafel slope higher than 200 mV dec⁻¹) might be limited by the formation of OH*, due to the unfavorable water dissociation to form Co^{III}-OH species on the surface (Figure 4b).”

At last, we thank the Reviewer for your help in evaluating the revised manuscript!

REVIEWERS' COMMENTS

Reviewer #1 (Remarks to the Author):

Authors responded to my questions enough, and the manuscript has been dramatically improved. Thus, I agree with the acceptance of this manuscript to “Nature Communications”.

Reviewer #2 (Remarks to the Author):

My questions have been well addressed.